# The Interplay of Epigenetic, Genetic, and Traditional Risk Factors on Blood Pressure: Findings from the Health and Retirement Study

**DOI:** 10.3390/genes13111959

**Published:** 2022-10-27

**Authors:** Xinman Zhang, Farah Ammous, Lisha Lin, Scott M. Ratliff, Erin B. Ware, Jessica D. Faul, Wei Zhao, Sharon L. R. Kardia, Jennifer A. Smith

**Affiliations:** 1Department of Epidemiology, School of Public Health, University of Michigan, Ann Arbor, MI 48109, USA; 2Survey Research Center, Institute for Social Research, University of Michigan, Ann Arbor, MI 48104, USA

**Keywords:** methylation risk score, DNA methylation, genetic risk score, interaction, genetics, blood pressure

## Abstract

The epigenome likely interacts with traditional and genetic risk factors to influence blood pressure. We evaluated whether 13 previously reported DNA methylation sites (CpGs) are associated with systolic (SBP) or diastolic (DBP) blood pressure, both individually and aggregated into methylation risk scores (MRS), in 3070 participants (including 437 African ancestry (AA) and 2021 European ancestry (EA), mean age = 70.5 years) from the Health and Retirement Study. Nine CpGs were at least nominally associated with SBP and/or DBP after adjusting for traditional hypertension risk factors (*p* < 0.05). MRS_SBP_ was positively associated with SBP in the full sample (β = 1.7 mmHg per 1 standard deviation in MRS_SBP_; *p* = 2.7 × 10^−5^) and in EA (β = 1.6; *p* = 0.001), and MRS_DBP_ with DBP in the full sample (β = 1.1; *p* = 1.8 × 10^−6^), EA (β = 1.1; *p* = 7.2 × 10^−5^), and AA (β = 1.4; *p* = 0.03). The MRS and BP-genetic risk scores were independently associated with blood pressure in EA. The effects of both MRSs were weaker with increased age (*p*_interaction_ < 0.01), and the effect of MRS_DBP_ was higher among individuals with at least some college education (*p*_interaction_ = 0.02). In AA, increasing MRS_SBP_ was associated with higher SBP in females only (*p*_interaction_ = 0.01). Our work shows that MRS is a potential biomarker of blood pressure that may be modified by traditional hypertension risk factors.

## 1. Introduction

Hypertension is a complex polygenic disorder with both genetic and environmental factors influencing blood pressure (BP). It is a major risk factor for cardiovascular disease and a leading cause of death globally [1,2]. In the United States, nearly half of the adult population has hypertension according to prevalence estimates from the National Health and Nutrition Examination Survey (NHANES, 2017–2018) [3]. Hypertension is more prevalent among non-Hispanic Blacks compared to non-Hispanic Whites and other racial/ethnic minorities [3,4]. In addition to genetic predisposition, older age, and male sex, several other exposures increase BP and hypertension risk including lower socioeconomic status, behavioral factors (e.g., poor diet, lack of exercise, smoking), and comorbidities (e.g., type 2 diabetes, chronic kidney disease) [5]. A full understanding of hypertension risk and etiology necessitates a consideration of genetic and non-genetic factors and their interactions.

Epigenetic mechanisms, including DNA methylation, dynamically regulate transcription and act as a molecular link between environmental stimuli and gene expression. A growing body of evidence suggests that DNA methylation levels at specific cytosine-phosphate-guanine (CpG) sites are associated with hypertension [6,7] and related risk factors and co-morbidities such as obesity [8,9] and type 2 diabetes [10,11,12]. Evidence of differential DNA methylation has also been reported for lifestyle and behavioral factors that can influence BP, including smoking [13,14,15,16], exercise [17], and stress [18,19]. Moreover, the epigenome may be a mediator of socioeconomic and neighborhood influences on health [20]. Previous studies have also shown that epigenetic changes are associated with educational attainment and income [21,22,23,24], neighborhood disadvantage [25,26] and perceived discrimination [27,28]. Hence, the epigenome may be a valuable biomarker of environmental stimuli although our understanding of the specific interplay between epigenetic, genetic, and traditional risk factors on BP remains limited.

The largest epigenome-wide study of BP to date, which included approximately 17,000 participants of European ancestry (EA), African ancestry (AA), and Hispanic ethnicity from the Cohorts for Heart and Aging Research in Genomic Epidemiology (CHARGE) Consortium, identified 13 CpG sites significantly associated with systolic (SBP) or diastolic (DBP) blood pressure [6]. In this work, we evaluated the association between these 13 CpGs, both individually and aggregated into methylation risk scores (MRS_SBP_ and MRS_DBP_), and SBP or DBP in participants from the Health and Retirement Study (HRS). We also examined how ancestry-specific BP genetic risk scores (GRS) and traditional risk factors including age, sex, and educational attainment interact with the MRSs to influence BP. To elucidate ancestry-specific effects, we performed analyses both in the full HRS sample as well as in EA and AA samples only.

## 2. Materials and Methods

### 2.1. Study Sample

The Health and Retirement Study (HRS) is a longitudinal panel study that is nationally representative of participants over age 50 in the United States. Details of the HRS sample and methods have been previously described [29]. Briefly, every two years, half of the HRS sample is interviewed by telephone and the other half has an enhanced face-to-face interview that includes blood pressure measurement. The following wave, the half-samples are interviewed using the alternative mode (i.e., each participant has a face-to-face interview every four years). Additional collection efforts were completed to obtain saliva for genotyping (2006–2012) and venous blood draws (Venous Blood Study (VBS), 2016) for DNA methylation and other biomarkers.

A total of 3560 participants had DNA methylation and genotype data. We excluded participants who were at least a 2nd degree relative to another study participant (*n* = 30), had a high missing call rate for genetic data (*n* = 2), were not eligible for physical examination (nursing home residents or proxy interviews) (*n* = 302), did not provide consent for physical assessment (*n* = 83), were missing BP measurements (*n* = 64), or were missing antihypertensive medication use information (*n* = 11). The final full sample included 3070 participants (Figure 1). A combination of self-reported race (White, Black, or other), self-reported Hispanic ethnicity, and genetic ancestry was used to identify ancestrally homogenous analytic samples of African ancestry (AA, *n* = 437) and European ancestry (EA, *n* = 2021) [30]. Our full sample also included 612 participants who self-reported their race/ethnicity as Hispanic (*n* = 416), non-Hispanic White (*n* = 73), non-Hispanic Black (*n* = 30), and other (*n* = 93). 

### 2.2. Blood Pressure 

Blood pressure was measured using the Omron HEM-780 Intellisense Automated blood pressure monitor (Omron Healthcare, Bannockburn, IL, USA) during face-to-face interviews in 2016 (*n* = 1655) or 2018 (*n* = 1415) [31]. Three measurements were taken 45 s apart on the participant’s left arm. Measurements of SBP > 250 mmHg and DBP < 40 mmHg were censored (18 BP measurements for 18 participants), and the participant’s blood pressure was calculated as the mean of the remaining BP measurements. For participants using antihypertensive medications, we added 15 mmHg and 10 mmHg to the average SBP and DBP, respectively [6].

### 2.3. Covariates

Sex was self-reported at the baseline HRS interview for each participant. Educational attainment was categorized as a three-level variable (less than high school, high school degree or equivalent, and at least some college) for all analyses except for those examining interactions. For interaction analysis, two dichotomous variables were used to indicate whether a participant attained a high school degree or equivalent (yes/no) and whether they had at least some college education (yes/no). Parental education was collected as years of education (continuous) except in the HRS Asset and Health Dynamics Among the Oldest Old (AHEAD) cohort which categorized parental education as <8 years or 8+ years. For this analysis, we dichotomized parental educational attainment as <12 years reported for all parents or ≥ 12 years for at least one parent [32]. For the AHEAD cohort, parental education < 8 years was included in the group with <12 years, but those with 8+ years of parental education were set to missing. Body mass index (BMI) was defined as the participant’s measured weight (kg) divided by height (m^2^). Type 2 diabetes was defined as self-reported use of oral hypoglycemics or insulin or having a blood glucose level ≥ 126 mg/dL. Alcohol consumption was categorized as nondrinker, occasional drinker (≤1 drink/day for females and ≤2 drinks/day for males), and heavy drinker (>1 drink/day for females and >2 drinks/day for males). Smoking status was categorized as never, former, or current smoker. Physical exercise was defined as engaging in moderate or vigorous intensity activity for 2–3 times a week versus not. Information on comorbidities and health behaviors were obtained from the concurrent BP measurement wave, either 2016 or 2018, except for type 2 diabetes which was based on glucose measures collected in 2016.

### 2.4. DNA Methylation Data and Methylation Risk Scores

DNA methylation was measured using the Infinium Methylation EPIC BeadChip from whole blood samples collected for 4104 individuals who participated in the 2016 VBS. Samples were randomized across plates by key demographic variables including age, cohort, sex, educational attainment, and race/ethnicity, with 40 pairs of blinded duplicates [33]. Data preprocessing and quality control were performed using the *minfi* package in R. Probes with a detection *p* > 0.01 were removed, and samples with >5% missing probes were removed [33]. White blood cell counts were estimated using the unconstrained option in the *ewastools* R package based on the Houseman algorithm and Salas reference panel [34,35]. After further excluding sex mismatches, DNA methylation data was available for 4018 participants. DNA methylation at each site, or CpG, was quantified using beta values calculated as the ratio of methylated probe intensity to the overall intensity (sum of methylated and unmethylated probes intensities). 

In this analysis, we focused on the 13 CpGs previously associated with SBP and DBP in an epigenome-wide association study (EWAS) of 17,010 participants of European (*n* = 11,345), African (*n* = 4636), and Hispanic (*n* = 1029) ancestries/ethnicities [6]. Using linear mixed modeling, the DNA methylation level at each CpG was first adjusted for white blood cell counts (fixed effects), chip (random effect), and slide (random effect). We then constructed two MRSs, MRS_SBP_ and MRS_DBP_, by summing the adjusted methylation levels at each CpG weighted by the effect sizes reported in the EWAS [6]. That is, MRS_SBP_ and MRS_DBP_ included the same 13 CpGs, but with different weights. Further, since the EWAS was conducted using a multi-ancestry sample, the MRSs for EA and AA were calculated using identical sets of CpG weights. MRSs were coded such that a larger MRS reflects lower overall methylation and higher BP. For each MRS, outliers beyond five standard deviations (SDs) from the mean were excluded (*n* = 3 for MRS_SBP_ and *n* = 3 for MRS_DBP_). All MRSs were standardized to a standard normal distribution. 

### 2.5. Genotype Data and Genetic Risk Scores 

Participants who consented to saliva collection between 2006 and 2012 were genotyped using Illumina Human Omni2.5 arrays (Illumina Inc., San Diego, CA, USA). After quality control, there were a total of 18,916 HRS participants with genotype data [30]. Principal components (PCs) of genotype data in unrelated study samples and 1118 HapMap controls were calculated using 155,322 single nucleotide polymorphisms (SNPs) selected by linkage disequilibrium (LD) pruning (r^2^ < 0.1 in a sliding 10 Mb window) [30]. The EA analytic sample included all self-reported non-Hispanic White participants that were within one SD of the mean of the first eigenvector of the PC analysis. The AA sample included all self-reported non-Hispanic Black participants within two SDs of the mean of the first eigenvector and within one SD of the mean for the second eigenvector. PCs were then recalculated within the full HRS genetic sample alone, and this set of PCs were used for analyses including GRSs conducted in our full analytic sample. Next, ancestry-specific PCs were calculated in the EA and AA analytic samples separately, which were used in our ancestry-specific analyses that included GRSs.

GRSs for SBP and DBP were constructed separately for EA and AA using SNP weights from the largest and most recent genome-wide association studies (GWAS) conducted within each ancestry group. The GRS in AA was constructed using summary statistics from the 2017 Continental Origins and Genetic Epidemiology Network-Blood Pressure (COGENT-BP) study, consisting of 31,968 individuals of African ancestry and validated with additional 54,395 individuals from multiple ancestries [36]. Since HRS AA participants were included in the COGENT-BP study, we used summary statistics re-calculated without HRS participants. The GRS in EA was constructed using summary statistics from the combined meta-analysis of participants from the UK Biobank and the International Consortium of Blood Pressure (ICBP) GWAS which included over 750,000 participants of European ancestry [37]. For both AA and EA, the GRSs included all available directly genotyped (not imputed) SNPs, weighted by the effect estimates from the respective ancestry-specific GWASs [38]. The decision to include all genotyped SNPs in the GRSs released by HRS was made after extensive analysis comparing multiple thresholds for clumping/pruning and significance thresholds for a variety of traits. Overall, for most traits in HRS, including all genotyped SNPs without any LD pruning performed similarly to other methods [38]. All GRSs were standardized to a standard normal distribution within ancestry.

### 2.6. Statistical Analysis

Frequencies and means of the sample characteristics were calculated in the full sample and by ancestry. Differences by ancestry were evaluated using two sample *t*-tests or Chi-square tests. Pearson’s correlation coefficients (r) were used to examine correlation between the MRSs and GRSs.

Linear regression models were used to assess the associations between DNA methylation (CpGs and MRSs) and SBP or DBP in the full sample and stratified by genetic ancestry (EA and AA). The effect estimates for the associations between the individual CpGs and BP were scaled to correspond to mmHg changes in BP for every 1% increase in the DNA methylation beta value. In all models, we tested the MRS association with the corresponding BP measure only (e.g., MRS_SBP_ association with SBP, not DBP). Model 1 was adjusted for age, sex, and the first 10 genetic PCs for each sample. Model 2 also included educational attainment, parental education, BMI, type 2 diabetes, alcohol consumption, smoking status, and physical exercise. We then added each BP-GRS to the ancestry-stratified analysis. For both SPB and DBP, we reported the percent of variance explained as the adjusted R^2^ from models without either the MRS or GRS (covariates only), and compared to models that include the MRS, GRS, or MRS + GRS. For individual CpGs, replication of the previous EWAS [6] in HRS was defined as a consistent effect direction and *p* < 0.05. We also applied Bonferroni correction to account for testing 13 CpGs with each BP measure, although we note that this approach is likely to be overly conservative due to correlation among the CpGs. In light of this, we noted both CpGs that were nominally significant (*p* < 0.05) and those that were significant after accounting for 13 tests (*p* < 0.0038). For all other associations, *p* < 0.05 was considered significant. We also used Cochran’s Q test to investigate whether the effects of the MRSs on BP were different between EA and AA.

Next, we assessed whether the GRSs, age, sex, having earned a high school degree, or having at least some college education modified the effect of the MRSs on BP by including the respective multiplicative interaction terms in the linear regression model. For significant interactions, we visualized the associations by plotting the predicted BP values by the MRS at each level of the categorical risk factors (sex and educational attainment) and at the 25th and 75th percentile values for continuous risk factors (age and GRSs). *p* < 0.05 was considered significant. Analyses were conducted in R using the *lme4*, *emmeans*, *sjPlot*, and *meta* packages. 

The HRS methylation sample is not a random sub-sample of HRS, and it is not by itself representative of any specific population. To evaluate our inferences in the U.S. population, we conducted a sensitivity analysis by including the HRS sampling weights that are released with the DNA methylation data for participants aged 55 and older. Including these weights allowed us to leverage the complex HRS study design to approximate a sample that is nationally representative of the U.S. population over the age of 55. A total of 2607 participants (342 AA, 1772 EA, and 493 others) were included in the sensitivity analysis that examined the associations between MRSs, GRSs, and MRS-by-risk factor interactions with BP using the models described in the primary analysis. Sensitivity analyses were conducted using the *survey* R package [39]. 

## 3. Results

### 3.1. Characteristics of Study Participants

The full study sample included 3070 HRS participants (437 of African ancestry, 2021 of European ancestry, and 612 of other ancestries) (Table 1). The sample was 58.5% female and the mean age at the time of BP measurement was 70.5 (SD = 9.5) years. Over half of the sample reported anti-hypertensive medication use (57.7%) and 28.2% had type 2 diabetes. About a third of the participants reported occasional drinking and 6% reported heavy drinking. About 10% of the participants were current smokers and about half reported weekly physical exercise. All sample characteristics differed significantly by ancestry (EA vs. AA) at *p* < 0.05. Overall, EA participants were older and had a higher educational attainment. Mean SBP and DBP were higher in AA participants, and they were more likely to have type 2 diabetes and higher mean BMI. The AA participants had a higher percentage of current smoking and no weekly physical exercise. A higher percentage of EA participants reported heavy or occasional drinking. Histograms of the overall and ancestry-specific MRSs are shown in Appendix A.

### 3.2. Associations between the 13 Previously Identified CpGs and Blood Pressure

The associations between the 13 previously identified BP-associated CpGs with SBP and DBP are shown in Table 2 for the full sample and in Appendix A for the EA and AA samples. Ten of the CpGs were at least nominally associated (*p* < 0.05) with BP after adjusting for age, sex, and genetic PCs (Model 1) for both SBP and DBP in the full sample, and remained significant after Bonferroni correction for 13 tests. Five (SBP) and seven (DBP) CpGs were nominally significant after further adjusting for educational attainment, parental education, BMI, type 2, diabetes, alcohol consumption, smoking status, and physical exercise (Model 2), although the magnitude of effect was attenuated in most cases. Two CpGs, cg00533891 and cg02711608, remained significant in Model 2 for DBP after Bonferroni correction. For all associations, there was an inverse relationship between DNA methylation and BP, analogous to the directions reported by the CHARGE consortium [6]. 

In AA, two CpGs were at least nominally associated with SBP in Model 1 only, while in EA, seven CpGs were associated. Five of the associated CpGs from Model 1 in EA remained associated after further adjustment for covariates in Model 2, while none remained significant in AA (*p* < 0.05) (Appendix A). In the AA DBP analyses, two CpGs were nominally associated in Model 1, one of which remained associated in Model 2, but not after Bonferroni correction (Appendix A). In EA, a total of ten CpGs were nominally associated in Model 1, with half of them significant after Bonferroni correction. In Model 2, six CpGs were nominally significant for DBP and cg02711608 remained significant after Bonferroni correction. For several of the CpGs that were associated in EA only, the effect estimates were comparable in AA but were not significant, likely due to the smaller sample size in AA. Three CpGs, cg06690548 (located in the gene body of *SLC7A11*), cg00533891 (located in the 5′UTR of *ZMIZ1*), and cg02711608 (located in the 1st exon, 5′UTR, or gene body of *SLC1A5* transcripts), were associated with both SBP and DBP in the full and EA samples (Model 2). One CpG, cg00533891, was associated with both SBP and DBP in AA and remained associated in Model 2 for DBP. 

### 3.3. Associations between the Methylation and Genetic Risk Scores and Blood Pressure

Next, we evaluated whether MRS_SBP_ and MRS_DBP_ were associated with BP in HRS (Table 3). Higher MRS_SBP_, indicative of lower overall DNA methylation, was associated with higher SBP in the full sample (*p* = 6.6 × 10^−9^), and in both AA and EA in Model 1 (*p* = 0.021 and *p* = 3.8 × 10^−6^, respectively). Adjusting for additional covariates in Model 2 slightly attenuated the associations in the full and EA samples and the association was no longer significant in the AA sample. The effect estimates were similar and ranged from 1.6 (AA and EA) to 1.7 (full sample) mmHg increase in SBP for each 1 SD increase in MRS_SBP_ in Model 2. MRS_DBP_ was associated with DBP in both the full and ancestry-specific Model 1 analysis (*p* = 9.3 × 10^−12^ for the full sample, *p* = 0.001 for AA, and *p* = 1.4 × 10^−8^ for EA). These associations attenuated in Model 2 but remained significant in all of the analytic samples with the effect estimates ranging from 1.1 (full sample and EA) to 1.4 (AA) mmHg increase in DBP for every 1 SD increase in MRS_DBP_. Although the MRS effect estimates differed somewhat between EA and AA, we found no evidence of effect heterogeneity in any of the Models using Cochran’s Q test (all *p* > 0.05).

The BP GRSs were constructed based on ancestry-specific estimates; hence we only evaluated their performance in ancestry-specific models. Overall, the GRSs were significant predictors of SBP and DBP in EA but not AA (Appendix A). In EA, a one SD increase in the GRS was associated with approximately 5 mmHg (*p* = 9.3 × 10^−16^) and 3 mmHg (*p* = 3.6 × 10^−16^) increase in SBP and DBP in Model 2, respectively. The MRSs and GRSs were weakly, but significantly, correlated with each other in EA (Pearson’s r = 0.05 for both SBP and DBP, *p* < 0.05), but were not correlated in AA. In AA, the correlation between the MRS and GRS was 0.02 (for DBP) and 0.03 (for SBP). Table 4 shows the BP associations for models that include both the MRSs and GRSs. In AA, MRS_DBP_ was significantly associated with DBP, but GRS_DBP_ was not. Neither MRS_SBP_ nor GRS_SBP_ was associated with SBP. In EA, both the MRSs and the GRSs were independently associated with their corresponding BP measure. When we examined the percentage of variance explained using the adjusted R^2^, models with both the MRS and GRS explained the highest percentage of variance in EA, which ranged between 9.5% for DBP and 12.1% for SBP (Table 5). This corresponded to 4% and 3.6% increases in the adjusted R^2^ compared to the model of covariates only and 0.6% and 0.4% increases compared to the covariates + GRS model for DBP and SBP, respectively. For AA, MRS_DBP_ did not explain additional variance compared to the model of covariates only. 

### 3.4. Interaction between Traditional Risk Factors and Methylation Risk Scores on Blood Pressure

Having identified MRSs as significant predictors of BP, we next evaluated whether their effects were modified by age, sex, educational attainment, and genetic risk. Table 6 shows the adjusted associations and interaction effects between the MRSs and risk factors in the full sample and by ancestry. 

We observed significant interaction between MRS_SBP_ and age in the full sample and EA, where the effects of MRS_SBP_ decreased at older age (β_interaction_ = −1.19, *p* = 0.002 for the full sample and β_interaction_ = −1.85, *p* = 7.0 × 10^−5^ for EA). Panels (a) and (b) in Figure 2 show the predicted SBP by MRS_SBP_ at the 25th and 75th percentile cutoff values of age for the full and EA samples. For the full sample, this was equivalent to 2.90 vs. 0.95 mmHg increases in SBP for each 1 SD in MRS_SBP_ at age 65.0 (25th percentile) and 77.5 (75th percentile), respectively. In EA, the corresponding effect estimates of MRS_SBP_ were 3.45 and 0.54 mmHg at age 64.0 (25th percentile) and 79.0 (75th percentile), respectively. We observed similar findings for MRS_DBP_ and age (β_interaction_ = −0.74, *p* = 0.001 for the full sample and β_interaction_ = −1.16, *p* = 1.9 × 10^−5^ for EA (panels (c) and (d) in Figure 2). For the full sample, this was equivalent to 1.86 vs. 0.65 mmHg increase in DBP for each 1 SD increase in MRS_DBP_ at age 65.0 (25th percentile) and 77.5 (75th percentile), respectively. In EA, the corresponding effect estimates of MRS_DBP_ were 2.22 and 0.40 mmHg at age 64.0 (25th percentile) and 79.0 (75th percentile). We noted a similar trend in AA, where the effects of the MRSs decreased with age (β_interaction_ = −0.46 for MRS_SBP_ and β_interaction_ = −0.48 for MRS_DBP_), but the associations did not reach statistical significance. 

In AA, we observed an interaction between MRS_SBP_ and sex on SBP, with the effect of MRS_SBP_ significant only in females (β_interaction_ = 6.36, *p* = 6.0 × 10^−3^, Figure 3). This was equivalent to 4.2 mmHg increase in SBP for each 1 SD increase in MRS_SBP_ in females compared to −2.16 mmHg in males. We noted similar effect gradients for MRS_DBP_ in AA, in addition to the full and EA samples (for both MRS_SBP_ and MRS_DBP_), where the effect estimates were higher in females compared to males, but these differences did not reach statistical significance. 

Having at least some college education had an interaction with MRS_DBP_ on DBP in which the effect of MRS_DBP_ was higher in the group with at least some college education in the full sample (β_interaction_ = 1.08, *p* = 0.02, Figure 4). This corresponded to 1.61 vs. 0.53 mmHg increase in DBP for each 1 SD increase in MRS_DBP_ among those with at least some college education vs. no college education, respectively. This trend was consistent but not statistically significant in the ancestry-specific analysis where the effects of both MRS_SBP_ and MRS_DBP_ were higher among participants with at least some college education. Finally, we did not observe significant interactions between any of the MRSs and the GRSs.

### 3.5. Sensitivity Analysis Using HRS Sampling Weights

Results of the sensitivity analysis using the HRS sampling weights to approximate a nationally representative sample of U.S. adults over age 55 are shown in Appendix A. Effect sizes for most of the significant associations from the primary analysis were similar or even larger in the sensitivity analysis; however, some of the associations were no longer significant, even at the nominal level, in the AA sample likely due to the reduced sample size. The effect sizes of the MRS associations with SBP and DBP were similar or larger compared to the analysis without weights (Appendix A). The associations of MRS_SBP_ with SBP (Model 1) and MRS_DBP_ with DBP (Model 2) in AA, however, were no longer significant. In models further including the GRS, MRS_DBP_ was no longer significantly associated with DBP in AA, while the remaining significant associations were generally larger in effect (Appendix A). The interaction between MRS_SBP_ and sex in AA was non-significant (*p* = 0.06) in the sensitivity analysis, but other interactions from the primary analysis remained significant (Appendix A). 

## 4. Discussion

This study characterized the association between DNA methylation, a link between environmental stimuli and gene expression, and BP in a multi-ancestral sample of older adults from the HRS. Most of the 13 previously identified BP-associated CpGs were at least nominally associated with SBP and DBP in the full sample of HRS. The epigenetic BP risk scores, MRS_SBP_ and MRS_DBP_, were associated with BP in the full and EA samples but not in the AA sample. The MRSs for SBP and DBP remained significant and independent predictors in models with their respective GRS in EA, accounting for 0.4% and 0.6% increase in the variance explained for SBP and DBP versus models with covariates and GRS, respectively. We also found that the MRSs had a larger effect on BP in younger participants and those with higher educational attainment in the full sample. Overall, our findings suggest that DNA methylation may be an important biomarker of blood pressure beyond traditional and genetic risk factors, at least in EA, although we note that the observed changes in BP associated with each standard deviation change in methylation risk score were relatively small (e.g., <1 to 5 mmHg).

Overall, the HRS sample was older and had higher SBP and DBP compared to participants from the CHARGE consortium analysis (means of combined CHARGE discovery and replication samples were 62.3 years, 132.6 mmHg for SBP, and 77.3 mmHg for DBP) [6]. Yet, we observed directions of effect consistent with the CHARGE analysis for most CpGs. In our models which further adjusted for alcohol consumption, type 2 diabetes, exercise, and educational attainment, we replicated associations in 9 of the 13 of the CpGs. Other studies have attempted to replicate findings from the CHARGE consortium and reported similar results. In one study of 4800 individuals of EA and AA ancestry (mean age range of 14 to 69), 8 of the 13 CpGs from the CHARGE consortium replicated, and seven of these overlapped with our findings [7]. In another analysis of approximately 1200 African Americans, three of the 13 CpGs replicated [40]. Of these, two overlapped with our findings from the full sample analysis, but not in AA. The replicated signal across these studies and the consistent direction of effect suggests that the associations between DNA methylation at a number of these CpGs and BP is robust and remains significant after accounting for traditional hypertension risk factors. 

The nominally significant CpGs identified in our study were also associated with a range of cardiometabolic phenotypes in previous EWAS including lipids [41,42], type 2 diabetes [12], and BMI [8,43,44], as well as alcohol consumption [45] and smoking [13]. This is not surprising given the known overlap between these conditions and/or risk factors with BP. For two of the significant CpGs in our study, cg06690548 and cg00574958, Richard and colleagues reported nearby changes in gene expression [6]. DNA methylation at cg06690548 was associated with increased expression of *SLC7A11* and increased blood pressure [6]. Solute carrier family 7 member 11 (*SLC7A11*) is a major regulator of metabolic reprogramming including nutrient dependency and intracellular redox balance. Increased expression *SLC7A11* was associated with poor prognosis in cancer [46], alcohol use disorders [47], and increased BP [6]. DNA methylation at cg00574958 was associated with decreased expression of *UNC93B1* and increased expression of *CPT1A. UNC93B1* encodes a protein that is involved in innate and adaptive immune response by regulating Toll-like receptor signaling. Carnitine palmitoyltransferase 1A (*CPT1A*) encodes an enzyme involved in the mitochondrial oxidation of long-chain fatty acids. DNA methylation at cg00574958 in the *CPT1A* gene region, was associated with lower risk of metabolic syndrome [48], lipids [48,49,50], BMI and waist circumference [51], and type 2 diabetes [11]. cg00533891, which was the only CpG associated with DBP in AA in our study, was associated with daytime SBP in a sample of Black participants [52], but was not associated with BP in a sample of about 1200 African Americans sibships [40]. CpG cg00533891 is in the *ZMIZ1* gene region which encodes a member of the PIAS (protein inhibitor of activated STAT) family of proteins that regulate the activity of various transcription factors, including the androgen receptor, Smad3/4, and p53, but has no reported BP association. 

In our study, risk scores based on DNA methylation were significant predictors of BP after accounting for traditional risk factors, including genetic risk scores, in individuals of European ancestry. The MRS likely captures time- and tissue-specific information that is distinct from genetic based estimates of risk. This suggests that an MRS for BP has the potential to further refine risk prediction and improve estimates of the explained phenotypic variance of BP. In our study, models with traditional risk factors explained between 8.5% (for SBP) and 5.5% (for DBP) in EA, and the MRSs explained approximately 1–2% of additional variance (1.8% in SBP, 0.7% in DBP). These MRSs based on the same 13 CpGs explained 1.4% more variance in SBP and 2.0% more variance in DBP compared to a model of age, sex, and BMI in participants from the Framingham Heart Study [6], which is comparable to our findings. In AA, MRS_DBP_ was significantly associated with DBP, but it did not explain additional variance beyond covariates as measured by adjusted R^2^. Our findings are in line with a previous work where we have shown that methylation risk scores can explain additional phenotypic variance compared to models of traditional risk factors for subclinical atherosclerosis [53]. McCartney and colleagues also reported similar findings for several complex phenotypes including BMI and lipids in which models including both epigenetic and polygenic scores explained more variance than models with either score alone [54]. 

In our study, the ancestry-specific BP GRSs were associated with BP measures in EA but not AA. This is likely due to several factors. First, the EA GWAS sample was over 20 times larger than the AA GWAS sample, allowing much more precise estimates of genetic effects for EA compared to AA. Second, the sample size of HRS AA was much smaller than EA, leading to reduced power to detect effects in AA. Further, recent evidence suggests that differences in genetic and environmental factors between populations of African ancestry may lead to reduced portability of GRSs across AA samples compared to other ancestries [55]. The MRS for BP, which was created from EWAS conducted in a multi-ancestry sample, was associated with both SBP and DBP in EA, but only DBP in AA. Lack of association for MRS_SBP_ in AA is likely a function of the small number of AA in our sample, since the effect estimate for MRS_SBP_ in AA was similar in magnitude to that in EA. Further studies are needed to evaluate the utility of MRSs in BP or hypertension prediction in individuals of non-European ancestries. Overall, our findings support the importance of genetic-based and epigenetic-based risk scores to understand BP variation at the individual level. 

Many studies have provided compelling evidence of differential DNA methylation in relation to aging [56,57,58] and cardiometabolic outcomes [12,54,59,60,61,62], but few studies have evaluated how a methylation-based risk score may interact with traditional risk factors. Our most noteworthy finding is the interaction between MRS and age, in which we detected a decreasing effect of both MRS_SBP_ and MRS_DBP_ on BP as age increased. Westerman and colleagues evaluated interactions between a methylation-based risk score and traditional risk factors in predicting cardiovascular disease [61]. They observed that the MRS was more predictive for those in lower “traditional” risk strata based on the Framingham risk score, which includes age. Both DNA methylation and BP have a strong association with age and further investigation of their interaction effects is needed. SBP is known to increase with age, while DBP tends to decrease with age mainly due to arterial stiffness, a trend that we also observe in our study. Both MRS_SBP_ and MRS_DBP_, however, show an interaction with age for BP where the effect attenuates as age increases. In addition to age, we also found interactions between MRSs and both sex and education in some HRS samples. The effect of MRS_SBP_ was significantly higher in females compared to males in AA, but this trend was not observed in EA. This observation should be interpreted with caution given that there were only 149 male AA included in this analysis. Our findings of inverse association between having at least some college education and DBP in the full sample were similar to previous studies [63,64,65], and we observed that the effect of MRS_DBP_ was stronger in participants with at least some college education. This may be in part due to environmental or behavioral factors having a greater impact on those with lower educational attainment. Further studies with larger sample sizes are needed to validate these findings. The lack of interaction between the GRS and MRS could suggest that each capture different genomic influences on BP. This is also supported by the small correlation between these scores that we observed in our sample. 

One limitation of our analysis is that BP measurements were obtained from two waves, one concurrent with DNA methylation assessment (2016) and the other two years later (2018). This was necessary since each HRS participant has BP measured only once every four years (i.e., for this sample, BP was measured either in 2016 or 2018, but not both). The reported associations are prone to reverse causation and further studies are needed to describe and delineate these associations longitudinally, especially in larger sample sizes of diverse ancestries. Our sample was comprised of older adults, and findings appear to generalize nationally to those aged 55 and older, but they may not generalize to younger individuals. Further, individuals included in our study may be healthier or are at lower risk of disease than the general population. A final limitation of our work is the lack of replication in other cohorts. However, our study provides a unique perspective on the role of the epigenome and its interaction with established risk factors on BP in individuals across multiple genetic ancestries. Our findings show that DNA methylation-based risk scores can be a biomarker of blood pressure after accounting for traditional risk factors. Further studies are needed to evaluate whether the MRSs are associated with hypertension-related outcomes, including cardiovascular diseases, and whether they may be leveraged toward improved prediction of risk in individuals of different ancestries. 

## Figures and Tables

**Figure 1 genes-13-01959-f001:**
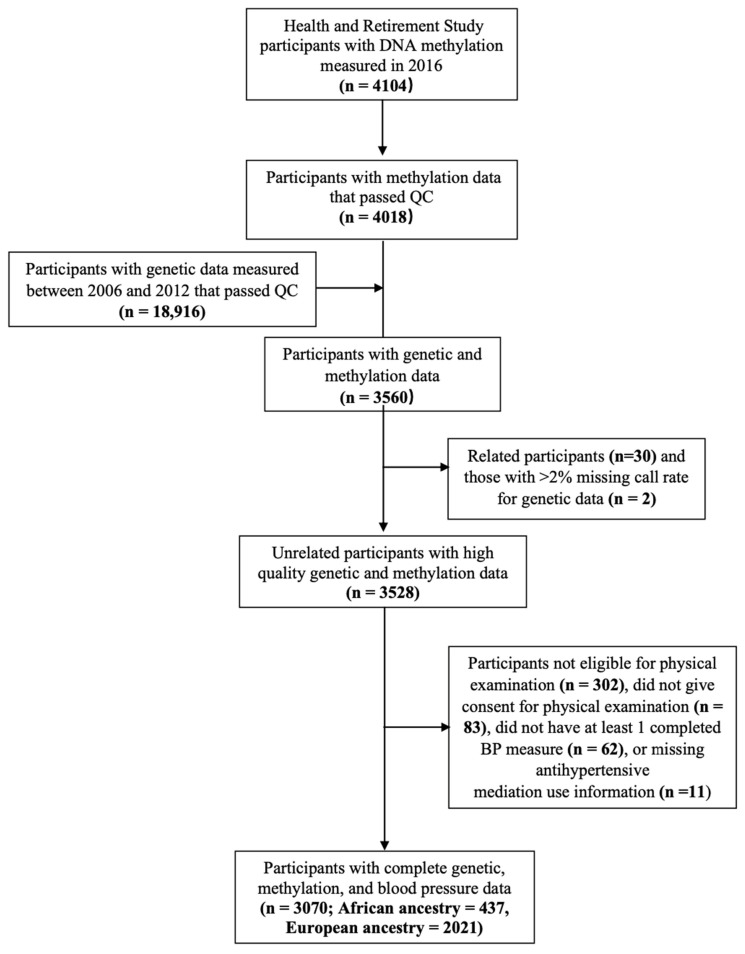
Study selection criteria for the Health and Retirement Study participants.

**Figure 2 genes-13-01959-f002:**
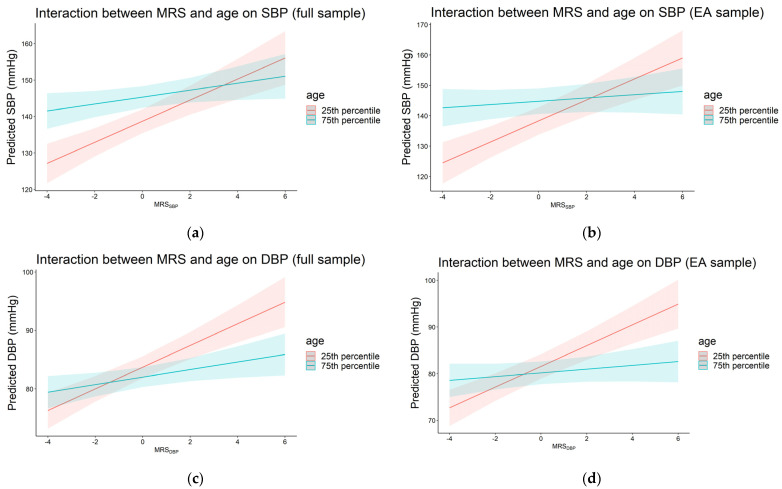
Plots of predicted systolic (**a**,**b**) and diastolic (**c**,**d**) blood pressure by MRS_SBP_ at the 25th and 75th percentiles of age in the full and European ancestry samples (*p*_interaction_ < 0.05). The 25th and 75th percentiles were equivalent to 65.0 and 77.5 years (full sample) and 64.0 and 79.0 years (European ancestry sample), respectively.

**Figure 3 genes-13-01959-f003:**
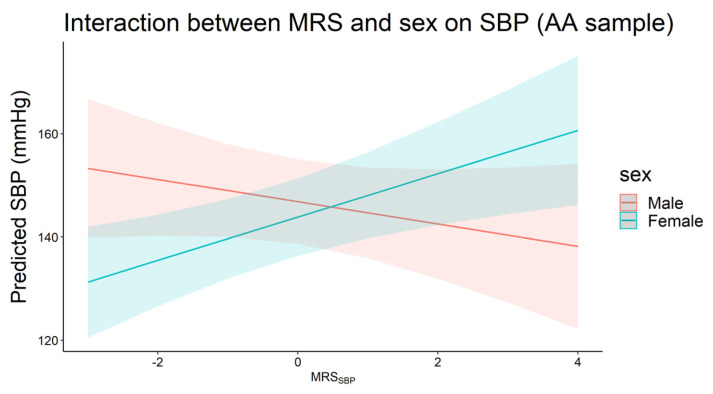
Plot of predicted systolic blood pressure by MRS_SBP_ for males (*n* = 149) and females (*n* = 288) in the African ancestry sample (*p*_interaction_ < 0.05).

**Figure 4 genes-13-01959-f004:**
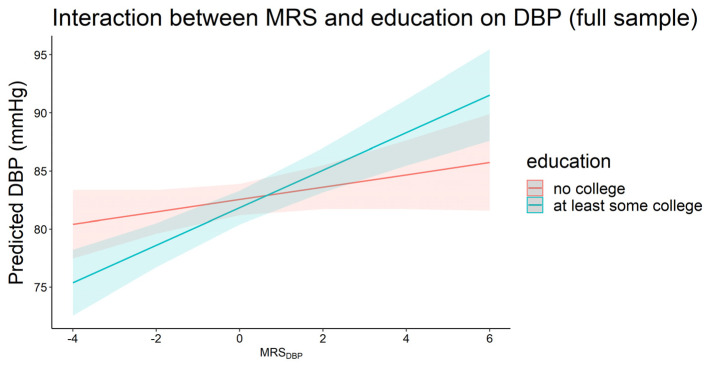
Plot of predicted diastolic blood pressure by MRS_DBP_ for those with (*n* = 1582) and without (*n =* 1488) at least some college education in the full sample (*p*_interaction_ < 0.05).

**Table 1 genes-13-01959-t001:** Descriptive characteristics of the Health and Retirement Study (HRS) sample.

	Full Sample ^a^ *n =* 3070 Mean (SD) or N (%)	African Ancestry *n =* 437 Mean (SD) or N (%)	European Ancestry *n* = 2021 Mean (SD) or N (%)	*p* ^b^
Age at blood pressure measurement (years)	70.5 (9.5)	67.3 (8.6)	72.3 (9.5)	<2.2 × 10^−16^
Male sex	1274 (41.5)	149 (34.1)	877 (43.4)	4.3 × 10^−4^
Educational attainment				<2.2 × 10^−16^
Less than high school	475 (15.5)	90 (20.6)	159 (7.9)	
High school degree or equivalent	1013 (33.0)	160 (36.6)	699 (34.6)	
At least some college	1582 (51.5)	187 (42.8)	1163 (57.5)	
Highest parental education < 12 years	1130 (36.8)	210 (48.1)	568 (28.1)	<2.2 × 10^−16^
Missing	190 (6.2)	49 (11.2)	81 (4.0)	
BMI (kg/m^2^)	29.9 (6.2)	31.1 (6.6)	29.5 (6.1)	<2.2 × 10^−16^
Systolic blood pressure (mmHg)	136.6 (21.1)	142.9 (21.3)	135.3 (20.6)	2.0 × 10^−11^
Diastolic blood pressure (mmHg)	82.1 (11.9)	86.8 (12.3)	81.1 (11.6)	<2.20 × 10^−16^
Anti-hypertensive medication use	1770 (57.7)	323 (73.9)	1111 (55.0)	4.9 × 10^−13^
Has Type 2 diabetes	865 (28.2)	159 (36.4)	477 (23.6)	6.0 × 10^−8^
Missing	9 (0.3)	0	9 (0.5)	
Alcohol consumption				2.0 × 10^−3^
Nondrinker	1869 (60.9)	288 (65.9)	1161 (57.4)	
Occasional drinker	995 (32.4)	124 (28.4)	699 (34.6)	
Heavy drinker	196 (6.4)	21 (4.8)	156 (7.7)	
Missing	10 (0.3)	4 (0.9)	5 (0.2)	
Smoking status				2.1 × 10^−7^
Never smoker	1376 (44.8)	186 (42.6)	895 (44.3)	
Former smoker	1365 (44.5)	173 (39.6)	943 (46.7)	
Current smoker	328 (10.7)	78 (17.8)	182 (9.0)	
Missing	1 (0.03)	0	1 (0.05)	
Reported weekly physical exercise	1712 (55.8)	210 (48.1)	1184 (58.6)	2.9 × 10^−4^
Missing	16 (0.5)	2 (0.5)	9 (0.5)	

SD, standard deviation; BMI, body mass index. ^a^ The overall sample includes the African ancestry sample, the European ancestry sample, and 612 additional participants who self-reported their race/ethnicity as Hispanic (*n* = 416), non-Hispanic White (*n* = 73), non-Hispanic Black (*n* = 30), and other (N = 93). ^b^ *p* represents the *p*-value for test of significant differences between AA and EA.

**Table 2 genes-13-01959-t002:** Associations between 13 blood pressure-associated CpGs and systolic or diastolic blood pressure in full sample of the Health and Retirement Study.

		Systolic Blood Pressure						Diastolic Blood Pressure					
		Model 1 (*n* = 3070)			Model 2 (*n* = 2694)			Model 1 (*n* = 3070)			Model 2 (*n* = 2694)		
CpG Site	UCSC Gene and Location ^b^	β	SE	*p*	β	SE	*p*	β	SE	*p*	β	SE	*p*
cg23999170	*TSPAN2* (Body)	−0.04	0.06	0.49	−0.04	0.07	0.51	−0.04	0.04	0.25	−0.05	0.04	0.15
cg16246545	*PHGDH* (Body)	**−0.18**	**0.06**	**5.0 × 10^−3^**	**−0.14**	**0.07**	**0.04**	**−0.10**	**0.04**	**0.01**	−0.07	0.04	0.08
cg14476101	*PHGDH* (Body)	**−0.15**	**0.05**	**3.3 × 10^−3 a^**	−0.10	0.05	0.06	**−0.10**	**0.03**	**7.3 × 10^−4 a^**	**−0.06**	**0.03**	**0.05**
cg19693031	*TXNIP* (3′UTR)	**−0.24**	**0.06**	**4.6 × 10^−5 a^**	**−0.14**	**0.06**	**0.03**	**−0.08**	**0.03**	**0.02**	−0.03	0.04	0.42
cg08035323	*–*	−0.05	0.05	0.33	−0.08	0.06	0.15	−0.03	0.03	0.25	−0.04	0.03	0.19
cg06690548	*SLC7A11* (Body)	**−0.27**	**0.07**	**5.6 × 10^−5 a^**	**−0.18**	**0.07**	**0.01**	**−0.17**	**0.04**	**6.4 × 10^−6 a^**	**−0.11**	**0.04**	**0.01**
cg18120259	*LOC100132354* (Body)	**−0.26**	**0.09**	**0.01**	−0.12	0.1	0.22	**−0.18**	**0.05**	**4.6 × 10^−4 a^**	**−0.12**	**0.05**	**0.03**
cg00533891	*ZMIZ1* (5′UTR)	**−0.16**	**0.06**	**0.01**	**−0.18**	**0.07**	**0.01**	**−0.13**	**0.04**	**4.3 × 10^−4 a^**	**−0.13**	**0.04**	**4.7 × 10^−4 a^**
cg17061862	*–*	**−0.15**	**0.06**	**0.01**	−0.11	0.06	0.05	**−0.10**	**0.03**	**1.6 × 10^−3 a^**	**−0.07**	**0.03**	**0.03**
cg00574958	*CPT1A* (5′UTR)	**−0.86**	**0.22**	**7.3 × 10^−5 a^**	−0.40	0.23	0.09	**−0.51**	**0.12**	**3.9 × 10^−5 a^**	−0.17	0.13	0.21
cg10601624	*–*	−0.10	0.09	0.28	−0.12	0.10	0.23	−0.08	0.05	0.15	−0.07	0.06	0.19
cg22304262	*SLC1A5* (5′UTR; Body)	**−0.24**	**0.07**	**1.2 × 10^−3^**	−0.10	0.08	0.20	**−0.16**	**0.04**	**8.3 × 10^−5 a^**	**−0.11**	**0.04**	**0.01**
cg02711608	*SLC1A5* (1st Exon; 5′UTR; Body)	**−0.53**	**0.11**	**2.5 × 10^−6 a^**	**−0.32**	**0.12**	**0.01**	**−0.34**	**0.06**	**1.2 × 10^−7 a^**	**−0.23**	**0.07**	**6.9 × 10^−4 a^**

SE, standard error; Chr, chromosome; SBP, systolic blood pressure; DBP, diastolic blood pressure. Model 1: SBP/DBP~ CpG site + age + sex + 10 PCs. Model 2: SBP/DBP~Model 1 covariates + smoking status + alcohol consumption + BMI + exercise + type 2 diabetes + educational attainment + parental education. Effect sizes (β) correspond to the change in SBP/DBP (mmHg) associated with a 1% increase in DNA methylation beta value of the CpG. Associations significant at *p* < 0.05 are shown in bold. ^a^ Significant after Bonferroni correction for 13 tests (*p* < 0.0038). ^b^ From Illumina annotation as reported in [6].

**Table 3 genes-13-01959-t003:** Associations between methylation risk scores and blood pressure in the full sample and by ancestry in the Health and Retirement Study.

	Systolic Blood Pressure								Diastolic Blood Pressure							
	Model 1				Model 2				Model 1				Model 2			
	β	SE	*p*	*n*	β	SE	*p*	*n*	β	SE	*p*	N	β	SE	*p*	*n*
Full sample	**2.22**	**0.38**	**6.6 × 10^−9^**	3067	**1.71**	**0.40**	**2.68 × 10^−5^**	2691	**1.48**	**0.22**	**9.3 × 10^−12^**	3067	**1.12**	**0.23**	**1.8 × 10^−6^**	2691
African ancestry	**2.44**	**1.05**	**0.02**	437	1.64	1.14	0.15	355	**1.93**	**0.59**	**1.0 × 10^−3^**	437	**1.42**	**0.66**	**0.03**	355
European ancestry	**2.11**	**0.45**	**3.8 × 10^−6^**	2019	**1.64**	**0.48**	**1.0 × 10^−3^**	1811	**1.48**	**0.26**	**1.4 × 10^−8^**	2018	**1.11**	**0.28**	**7.2 × 10^−5^**	1810

SE, Standard Error; MRS, Methylation risk score; SBP, systolic blood pressure; DBP, diastolic blood pressure. Model 1: SBP/DBP~MRS_SBP_/MRS_DBP_ + age + sex + 10 PCs. Model 2: SBP/DBP ~ Model 1 covariates + smoking status + alcohol consumption + BMI + exercise + type 2 diabetes + educational attainment + parental education. Effect sizes (β) correspond to the change in SBP/DBP (mmHg) associated with a 1 standard deviation increase in the MRS. Associations significant at *p* < 0.05 are shown in bold.

**Table 4 genes-13-01959-t004:** Associations between methylation and genetic risk scores and blood pressure by ancestry in the Health and Retirement Study.

	Systolic Blood Pressure			Diastolic Blood Pressure		
Sample/Predictor	β	SE	*p*	β	SE	*p*
African ancestry (*n* = 355)						
MRS	1.59	1.14	0.17	**1.41**	**0.66**	**0.03**
GRS	−3.42	2.63	0.20	−0.80	1.31	0.54
European ancestry (*n* = 1811 for SBP and 1810 for DBP)						
MRS	**1.49**	**0.47**	**2.0 × 10^−3^**	**1.01**	**0.27**	**2.4 × 10^−4^**
GRS	**4.83**	**0.60**	**8.8 × 10^−16^**	**2.80**	**0.34**	**5.3 × 10^−16^**

MRS, methylation risk score; GRS, genetic risk score; SBP, systolic blood pressure; DBP, diastolic blood pressure. Model: SBP/DBP~MRS_SBP_/MRS_DBP_ + GRS_SBP_/GRS_DBP_ + age + sex + 10 PCs + smoking status + alcohol consumption + BMI + exercise + type 2 diabetes + educational attainment + parental education. MRS effect sizes (β) correspond to the change in SBP/DBP (mmHg) associated with a 1 standard deviation increase in the MRS. GRS effect sizes (β) correspond to the change in SBP/DBP (mmHg) associated with a 1 standard deviation increase in the GRS. Associations significant at *p* < 0.05 are shown in bold.

**Table 5 genes-13-01959-t005:** Adjusted R^2^ of systolic and diastolic blood pressure models by ancestry in the Health and Retirement Study.

Model	Systolic Blood Pressure	Diastolic Blood Pressure
African ancestry		
Covariates only (age, sex, 10 PCs, smoking status, alcohol consumption, BMI, exercise, type 2 diabetes, educational attainment, parental education)	2.9%	9.3%
Covariates + MRS	3.2%	8.9%
Covariates + GRS	3.1%	9.2%
Covariates + MRS + GRS	3.4%	10.1%
European ancestry		
Covariates (age, sex, 10 PCs, smoking status, alcohol consumption, BMI, exercise, type 2 diabetes, educational attainment, parental education)	8.5%	5.5%
Covariates + MRS	10.3%	6.2%
Covariates + GRS	11.7%	8.9%
Covariates + MRS + GRS	12.1%	9.5%

**Table 6 genes-13-01959-t006:** Interaction analysis between blood pressure methylation risk scores and risk factors by ancestry in the Health and Retirement Study.

		Systolic Blood Pressure							Diastolic Blood Pressure						
Multiplicative Interaction Term Evaluated	Sample	*n*	β_MRS_	P_MRS_	β_RF_	*p* _RF_	β_interaction_	*p* _interaction_	*n*	β_MRS_	*p* _MRS_	β_RF_	*p* _RF_	β_Interaction_	*p* _interaction_
MRS × GRS	AA	355	2.06	0.08	−3.77	0.15	−2.18	0.06	355	**1.67**	**0.01**	−1.06	0.42	−1.19	0.09
	EA	1811	**1.49**	**2.0 × 10^−3^**	**4.83**	**9.0 × 10^−16^**	−0.03	0.94	1810	**1.04**	**1.7 × 10^−4^**	**2.80**	**4.4 × 10^−16^**	−0.50	0.07
MRS × age ^a^	Full sample	2691	**1.83**	**6.9 × 10^−6^**	**4.07**	**1.1 × 10^−19^**	**−1.19**	**2.0 × 10^−3^**	2691	**1.19**	**3.6 × 10^−7^**	**−1.04**	**4.2 × 10^−5^**	**−0.74**	**1.0 × 10^−3^**
	AA	355	1.52	0.20	1.06	0.46	−0.46	0.70	355	1.29	0.06	**−2.30**	**6.0 × 10^−3^**	−0.48	0.50
	EA	1811	**2.19**	**1.1 × 10^−5^**	**4.11**	**9.5 × 10^−15^**	**−1.85**	**7.0 × 10^−5^**	1810	**1.43**	**7.1 × 10^−7^**	**−0.88**	**4.0 × 10^−3^**	**−1.16**	**1.9 × 10^−5^**
MRS × sex ^b^	Full sample	2691	**1.33**	**0.03**	**−3.23**	**7.4 × 10^−5^**	0.68	0.39	2691	**0.78**	**0.02**	−0.13	0.78	0.61	0.18
	AA	355	−2.16	0.23	−2.96	0.23	**6.36**	**6.0 × 10^−3^**	355	−0.01	0.99	−1.11	0.42	2.40	0.07
	EA	1811	**1.61**	**0.02**	**−2.44**	**0.01**	0.06	0.95	1810	0.77	0.05	0.94	0.09	0.65	0.23
MRS × high school degree or equivalent ^c^	Full sample	2691	0.57	0.61	**−3.36**	**0.01**	1.29	0.28	2691	0.81	0.21	−0.65	0.37	0.34	0.62
	AA	355	−0.53	0.85	−0.48	0.88	2.63	0.39	355	0.83	0.63	0.18	0.92	0.76	0.68
	EA	1811	−0.78	0.65	**−5.35**	**4.0 × 10^−3^**	2.59	0.15	1810	1.15	0.24	−1.20	0.26	−0.05	0.96
MRS × at least some college ^d^	Full sample	2691	1.00	0.09	**−1.85**	**0.03**	1.31	0.09	2691	0.53	0.11	−0.70	0.15	**1.08**	**0.02**
	AA	355	0.57	0.73	−2.43	0.31	1.97	0.37	355	0.51	0.59	−1.18	0.39	1.71	0.18
	EA	1811	1.06	0.13	−1.84	0.07	1.01	0.28	1810	0.74	0.07	−0.61	0.30	0.66	0.22

AA, African ancestry; EA, European ancestry; MRS, methylation risk score; GRS, genetic risk score; RF, risk factor; SBP, systolic blood pressure; DBP, diastolic blood pressure. Model: DBP/SBP ~ corresponding MRS/GRS + age + sex + 10 PCs + smoking status + alcohol consumption + BMI + exercise + type 2 diabetes + educational attainment + parental education + corresponding multiplicative interaction term (MRS × GRS/age/sex/high school degree/at least some college education). MRS effect sizes (β_MRS_) correspond to the change in SBP/DBP associated with a 1 standard deviation increase in the MRS. Risk factor effect sizes (β_RF_) correspond to the change in SBP/DBP associated with a 1 unit increase in the risk factor for continuous measures or at the non-reference level for categorical variables. Interaction effect sizes (β_interaction_) correspond to the change in effect of β_MRS_ on SBP/DBP for each 1 unit increase (or level) of the risk factor. Associations significant at *p* < 0.05 are shown in bold. ^a^ Age was centered and scaled. ^b^ Reference group = male sex. ^c^ Reference group = less than high school degree. ^d^ Reference group = no college education.

## Data Availability

HRS genetic and epigenetic data can be accessed via the National Institute on Aging Genetics of Alzheimer’s Disease Data Storage Site (NIAGADS, accession number: NG00119.v1). HRS genetic risk scores, blood pressure, and survey data are also publicly available from the HRS website at https://hrs.isr.umich.edu/data-products (accessed on 10 September 2021).

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
