# Peer review of "The Interplay of Epigenetic, Genetic, and Traditional Risk Factors on Blood Pressure: Findings from the Health and Retirement Study"

_genes, 2022, doi:10.3390/genes13111959_

Round 1
Reviewer 1 Report
The authors focus their MS on the role of DNA methylation in Blood Pressure. The study of Epigenetic modification, such as DNA methylation, is important in the evaluation of environmental factors on the genomic sequence. The authors use the Health and retirement study as the starting population where evaluate the importance of the interactions among DNA methylation and Blood Pressure. Using adequate methods, they show how the use of 13 already known CpG add information to the classically known genetic risk scores. The study has some limitations, transparently addressed by the authors, on the sample size of African Ancestry participants. This study adds information to what is already known. I would suggest to add, in the Discussion, some consideration on the small size of the impact of DNA methylation on Blood Pressure measurement (<1 to 5 mmHg)
Reviewer 2 Report
The authors present association analysis results from the Health & Retirement Study, investigating the association of previously reported CpGs and Blood Pressure, followed by aggregated Methylation Risk Scores (MRS) combining the effects of these CpGs together, and then investigating the interactions of these MRS effects with BP genetic risk scores and other traditional risk factors.
This is an original study to investigate together MRS, GRS and other risk factors all into one analysis for BP. The analyses have been performed well, and the paper is clear and interesting to read, with excellent presentation of results.
Comments or Suggestions:
1) The p-value of 0.05 used as replication threshold for the 13 individual CpG associations is not valid, without multiple testing correction.
2) More methods detail is required for the construction of the BP-GRS. The authors say “GRSs included all available genotyped SNPs…”. Is this definitely what the authors mean? If all genome-wide SNPs are used, then surely LD-pruning is required. Or are the authors only referring to the 901 SNPs reported in the Evangelou et al paper, used in the GRS by Evangelou et al? And similarly for the GRS for AA from the COGENT-BP-GWAS.
3) Regarding both the MRS and the GRS, when authors refer to ancestry-specific scores, please clarify whether this relates to different sets of SNPs, or simply different weights.
4) Please clarify, and potentially re-consider, as to whether the MRS include all 13 CpGs, or only the ones which were significant in the individual association analyses?
5) Could the heterogeneity in the results between EA vs AA be calculated?
6) I would apply caution to the comment “associations may differ by genetic ancestry” in the abstract, when there are substantial differences in statistical power with the much smaller sample size of AA vs EA.
7) Please could the authors comment more on why the BP-GRS are not significant in AA, as compared to EA. Clearly smaller N is the most likely reason. But I am not sure if the authors are also referring to differences in the GRS, either the SNPs in the GRS, or the SNP-weights, or the accuracy of the GWAS they originate from, etc. It is hard to ascertain this, given the lack of clarity in the BP-GRS methods, as per comment (2).
8) Similarly, please clarify comments on pg.10, regarding transferability of the GRS in different ancestries. If the GRS are constructed to be ancestry-specific, then I don’t understand why transferability can be an issue at play here…
9) On p.5, please correct the reference to the Evangelou et al paper, “…which included over 1 million participants…”, because the GWAS discovery of UKB + ICBP, from which the GRS is constructed, only had a discovery sample size of N~750k.
10) Please clarify “effect estimates re-calculated without the HRS contribution for the COGENT-BP-GRS”. Is this because HRS AA samples is one of the studies within the published COGENT meta-analysis?
11) From the bottom paragraph on p.10, please explain why BP measurements were pooled from the two 2016 & 2018 waves? Why did you need to do this? Was this to increase overall sample size of participants? Or are all subjects present at both 2016 & 2018, but BP data was used from both time-points? Please clarify. Are you able to do sensitivity analyses to compare 2016 vs 2018?
12) Yes, I agree with comments on p.10 that the interaction with sex should be treated with caution.
13) The only main limitation of the study is the lack of validation, with results only reported from one single HRS study.
14) Depending on Editor preferences, I would recommend Sup Tables S1 & 2 to be a Main Table, in a condensed/combined format. Similarly, I would recommend Table S4 as a Main Table, if space.
15) In Table S3, why is “MRS” in the legend, when the Table is only for GRS results? This is confusing.
